# Forecasting Road Traffic Deaths in Thailand: Applications of Time-Series, Curve Estimation, Multiple Linear Regression, and Path Analysis Models

**Sajjakaj Jomnonkwao [1],\* , Savalee Uttra [2] and Vatanavongs Ratanavaraha [1]**

[1]  School of Transportation Engineering, Institute of Engineering, Suranaree University of Technology, 111 University Avenue, Suranaree Sub-District, Muang District, Nakhon Ratchasima 30000, Thailand; vatanavongs@g.sut.ac.th

[2]  Department of Logistics Engineering and Transportation Technology, Faculty of Engineering and Industrial Technology, Kalasin University 62/1 Kaset Sombun Road, Kalasin Subdistrict, Mueang District, Kalasin 46000, Thailand; savalee.utt@gmail.com

\*   Correspondence: sajjakaj@g.sut.ac.th; Tel.: +66-4422-4251; Fax: +66-4422-4608

**Abstract:** In 2018, 19,931 people were killed in road accidents in Thailand. Thus, reduction in the number of accidents is urgently required. To provide a master plan for reducing the number of accidents, future forecast data are required. Thus, we aimed to identify the appropriate forecasting method. We considered four methods in this study: Time-series analysis, curve estimation, regression analysis, and path analysis. The data used in the analysis included death rate per 100,000 population, gross domestic product (GDP), the number of registered vehicles (motorcycles, cars, and trucks), and energy consumption of the transportation sector. The results show that the best three models, based on the mean absolute percentage error (MAPE), are the multiple linear regression model 3, time-series with exponential smoothing, and path analysis, with MAPE values of 6.4%, 8.1%, and 8.4%, respectively.

**Keywords:** accident forecasting; multiple linear regression model; time-series; path analysis

## 1. Introduction

### 1.1. Road Safety Situation in Thailand

The Decade of Action for Road Safety 2011–2020 initiative was introduced to increase awareness across various countries of the measures that lead to increased road safety. The United Nations (UN) General Assembly delegated the World Health Organization (WHO) to monitor the progress of the initiative through a series of documents, compiled into the Global Status Reports on Road Safety in 2015. This report described an estimated 1.25 million fatalities caused by accidents in 2013 and indicated that roads in low- and middle-income countries were less safe than those in high-income countries; in particular, the death rates were twice those of high-income countries. Thailand has been classified as a middle-income country and was second in the global ranking, with a death rate of 36.2 per 100,000 population (Libya ranked first with a rate of 73.4 per 100,000 population). Thailand has been reported to be the most dangerous country in the world for motorcycle users, with a death rate of 27.3 deaths per 100,000 population [1]. In November 2017, James Burton [2] reported that as Libyan officials had declared that the number of road fatalities in Libya was caused by fighting on the roads, not driving, Thailand was then ranked as having the global highest road-based death rate (Figure 1).

Reported that Thailand was first in the global rankings, followed by Malawi and Liberia, with death rates of 35.0 and 33.7 per 100,000 population, respectively. Similarly, the report World Health Statistics 2017: Monitoring Health for the Sustainable Development Goals (SDGs) identified Thailand as having the highest number of road deaths in the WHO Southeast Asia Region (SEAR); whereas, the Bolivarian Republic of Venezuela was identified as having the highest road death rate in the WHO Region of the Americas [3].

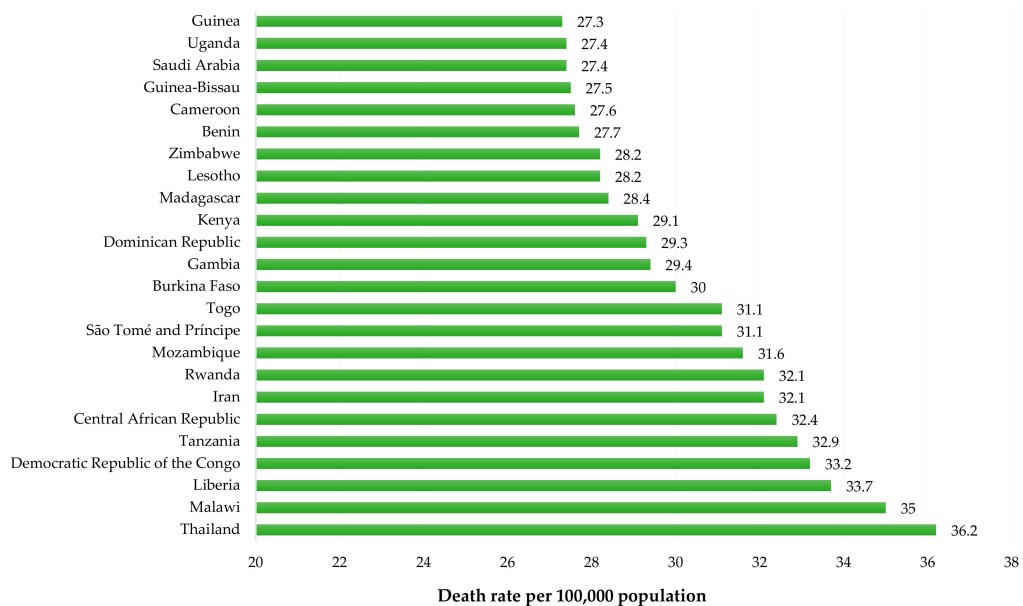

**Figure 1.** Statistics of worldwide death rates from road accidents per 100,000 population in 2013 James Burton [2].

In Thailand, road accident statistics are collected from many departments and have been regularly reported and organized in the form of annual reports. Each department has a different reason for recording individual data, which does not accurately reflect the situation in Thailand. According to the 2018 statistics, 19,931 people killed in road accidents in Thailand, as reported by the Integrated Information System for road deaths (RTDDI: Road Traffic Death Data Integration) [4], the Bureau of Non-Communicable Disease, the Department of Communicable Disease Control, and the Ministry of Public Health [5]. The system mentioned above had already begun integrating road accident fatality data from three databases in 2013, including the system for all deaths acquired from death notifications, including death certificates and death registration [5], data from the Royal Thai Police system (POLIS) consisting of case records [6], and data from the E-Claim system to compose the most comprehensive death data with the support of the Thai Health Promotion Foundation (ThaiHealth) and the WHO in Thailand [7].

### 1.2. Importance of Accident Prediction

Given these unsafe road situations, the reduction of accidents to reduce the number of deaths has become an urgent issue that must be addressed by the government. Forecasting the number of accidents on highways is necessary for developing road safety plans in terms of staff, budgets, and policies. In addition, instruments or techniques for effective forecasting must be found [8].

Examples of road safety policies in Thailand include law enforcement (e.g., for speed violations or alcohol consumption), road safety programs in educational institutions, creation of advertising media, an increase in the training hours required to obtain new driver's licenses or for renewals, engineering solution techniques for road safety audits, and research funding. To set these policies, the forecast data of the number of accidents have been used for determining the operating budgets.

### *1.3. Previous Study in Road Accident Prediction*

Various additional factors may potentially affect accident forecasting. According to international research studies (Table 1), time-series methods have been applied for accident prediction; for example, Quddus [9], Ramstedt [10], Dadashova et al. [11], García-Ferrer et al. [12], Zheng and Liu [13], Sanusi et al. [14], Parvareh et al. [15] and Dadashova et al. [16]. The Autoregressive integrated moving average (ARIMA) model has mostly been applied to time-series analysis.

Similar to regression analysis, Michalaki et al. [17] and Lu and Tolliver [18] used 6 and 18 years of historical data, respectively, for accident forecasting. Oh et al. [19], Garcia-Ferrer et al. [12], Zheng and Liu [13], and Ameen and Naji [20] used 4, 28, 13 and 17 years of historical data to predict accidents, respectively. Other techniques have also been applied, such as the Poisson regression model, the negative binomial model, the Conway–Maxwell–Poisson model, the Bernoulli model, the Hurdle Poisson model, the zero-inflated Poisson model (ZIP), and Neural network models.

Accident data include accident frequencies, the number of deaths, severe injuries, and minor injuries (including the number of pedestrian injuries caused by road accidents). Environmental data, such as location, time, weather, road conditions, crossing locations, infrastructure equipment, and historical accident information may be included in the analysis. For example, Lu and Tolliver [18] used the number of kilometers travelled by vehicles; Quddus [9] used vehicle characteristics and other behavioral factors; Michalaki et al. [17] used road pavement data; and the number of lanes were used by Oh et al. [19].

Economical factors have also been used in the analysis by Dadashova et al. [11]. García-Ferrer et al. [12] included energy consumption in their analysis. Alcohol consumption was considered by Ramstedt [10]. Some studies have included law enforcement in the analysis, e.g., Quddus [9]. Vehicle registration was used by García-Ferrer et al. [12] and Ameen and Naji [20]. Only Ameen and Naji [20] included population data in their analysis, along with other data such as, average daily traffic (ADT; vehicles per day) and other behavioral factors.

In general, most studies only applied one method, such as time-series analysis with an ARIMA model. If more than one method was used, it has typically been a regression model. Therefore, the study of other methods is necessary. Therefore, the aim of this study was to apply time-series analysis with 20 years of historical data from 1997 to 2016 using exponential smoothing and multiple linear regression techniques. Other techniques were applied: Curve estimation and path analysis—two methods that have not yet been used in any other study.

In our study, we included economic factors, energy consumption, vehicle registration, and population to develop our model. The effectiveness of the model was tested using the research question "What is an effective model for forecasting road traffic deaths?" Thus, we aimed to identify an appropriate method for forecasting road traffic deaths in Thailand and comparing the effectiveness of the models. Four methods were considered in this study: Time-series analysis, curve estimation, regression analysis, and path analysis. The remainder of this paper is structured as follows: The material and methods are presented in Section 2, followed by the results, research conclusions, and discussion, then the limitations and future work.

**Table 1.** Previous research on accident prediction and analytical methodologies.

| Author | Period | Methodology | | | Data | | | | | | | |
|---|---|---|---|---|---|---|---|---|---|---|---|---|
| | | Time-Series | Regression | Other | Accident Data | Environment Conditions | Economic Factors | Energy Consumption | Alcohol Consumption | Law | Vehicle Registration | Population |
| Quddus [9] | 1950–2005 (55 years) | ✓ | - | - | ✓ | ✓ | - | - | - | ✓ | - | - |
| Michalaki, et al. [17] | 2005–2011 (6 years) | - | ✓ | - | ✓ | ✓ | - | - | - | - | - | - |
| Lu and Tolliver [18] | 1996–2014 (18 years) | - | ✓ | ✓ | ✓ | ✓ | - | - | - | - | - | - |
| Ramstedt [10] | 1950–2002 (52 years) | ✓ | - | ✓ | ✓ | - | - | - | ✓ | - | - | - |
| Oh, et al. [19] | 1998–2002 (4 years) | - | ✓ | ✓ | ✓ | ✓ | - | - | - | - | - | - |
| Dadashova, et al. [11] | 2000–2011 (11 years) | ✓ | - | - | ✓ | ✓ | ✓ | - | - | ✓ | - | - |
| Dadashova, et al. [16] | 2000–2009 (9 years) | ✓ | - | ✓ | ✓ | ✓ | ✓ | - | - | ✓ | - | - |
| García-Ferrer, et al. [12] | 1975–2003 (28 years) | ✓ | ✓ | - | ✓ | - | ✓ | ✓ | - | - | ✓ | - |
| Zheng and Liu [13] | 1989–2002 (13 years) | ✓ | ✓ | ✓ | ✓ | - | - | - | - | - | - | - |
| Ameen and Naji [20] | 1978–1995 (17 years) | - | ✓ | - | ✓ | ✓ | ✓ | - | - | - | ✓ | ✓ |
| Sanusi, et al. [14] | 1969–2013 (54 years) | ✓ | - | - | ✓ | - | - | - | - | - | - | - |
| Parvareh, et al. [6,15] | 2009–2015 (72 months) | ✓ | - | - | ✓ | - | - | - | - | - | - | - |
| This research | 1997–2016 (20 years) | ✓ | ✓ | Curve estimate, Path analysis | ✓ | - | ✓ | ✓ | - | - | ✓ | ✓ |

## 2. Materials and Methods

### 2.1. Data Collection

In this research, we use Thailand's statistics for forecasting the death rate from road accidents. Thailand collects data from a wide range of departments and information from the integration of three databases; however, the number of years was not sufficient to develop the model. Other studies used historical data of the past 6 to 55 years for forecasting. Thailand has completely collected historical data for the past 20 years. Therefore, we used statistics of the past 20 years (1997–2016) of road accident fatalities according to the traffic lawsuit data from the Royal Thai Police [6].

The data used in our study included three data sets: Population data, economic data, and transportation data, including population statistics—obtained from the Department Provincial Administration [21]; gross domestic product (GDP)—obtained from the Bank of Thailand [22]; the number of registered vehicles—obtained from the Department of Land Transport [23]; the energy consumption of the Transport Sector—obtained from the Ministry of Energy [24].

### 2.2. Methodology

The analysis was divided into four parts: Studying past data trends; conducting the analysis using the developed model to forecast the death rates from road accidents using time-series analysis, curve estimation, multiple regression analysis, and path analysis; comparing forecast accuracy by seeking the optimal model with minimum errors; and using the model to make a prediction by forecasting trends for the next 10 years.

The statistical data trends for all components of our analysis are shown in Figure 2. Figure 2a depicts the death rate per 100,000 population; we discovered that the road death rate was 22.75 in 1997, decreasing to 20.89 in 2002. Then, it decreased steadily until 2010. However, the rate of road deaths increased over 10.34 in 2010 within a period of 3 years—from 11.33 in 2013 to 12.69 in 2016. The economic growth (as indicated by GDP) continuously grew from 4.71 billion baht in 1997 to 13.02 billion baht in 2016. The number of registered vehicles grew, starting from 17.67 million vehicles in 1997 and increasing to 37.34 million vehicles in 2016 (despite a decrease in 2004); the energy consumption for the transport sector tended to increase, from 20.25 Ktoe in 1997 to 30.19 Ktoe in 2016. This trend reflected Thailand's economic growth and improvements in the population well-being, as shown in Figure 2b–d, respectively.

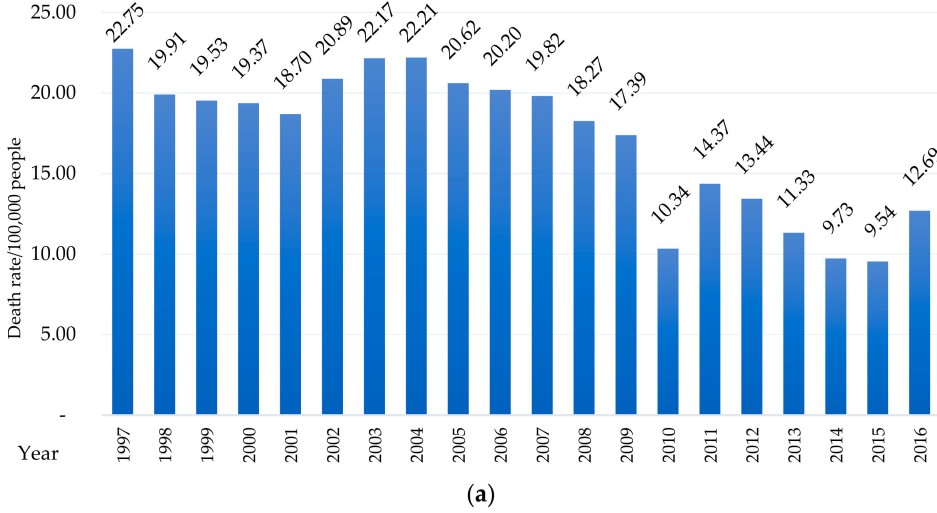

(a)

**Figure 2.** *Cont.*

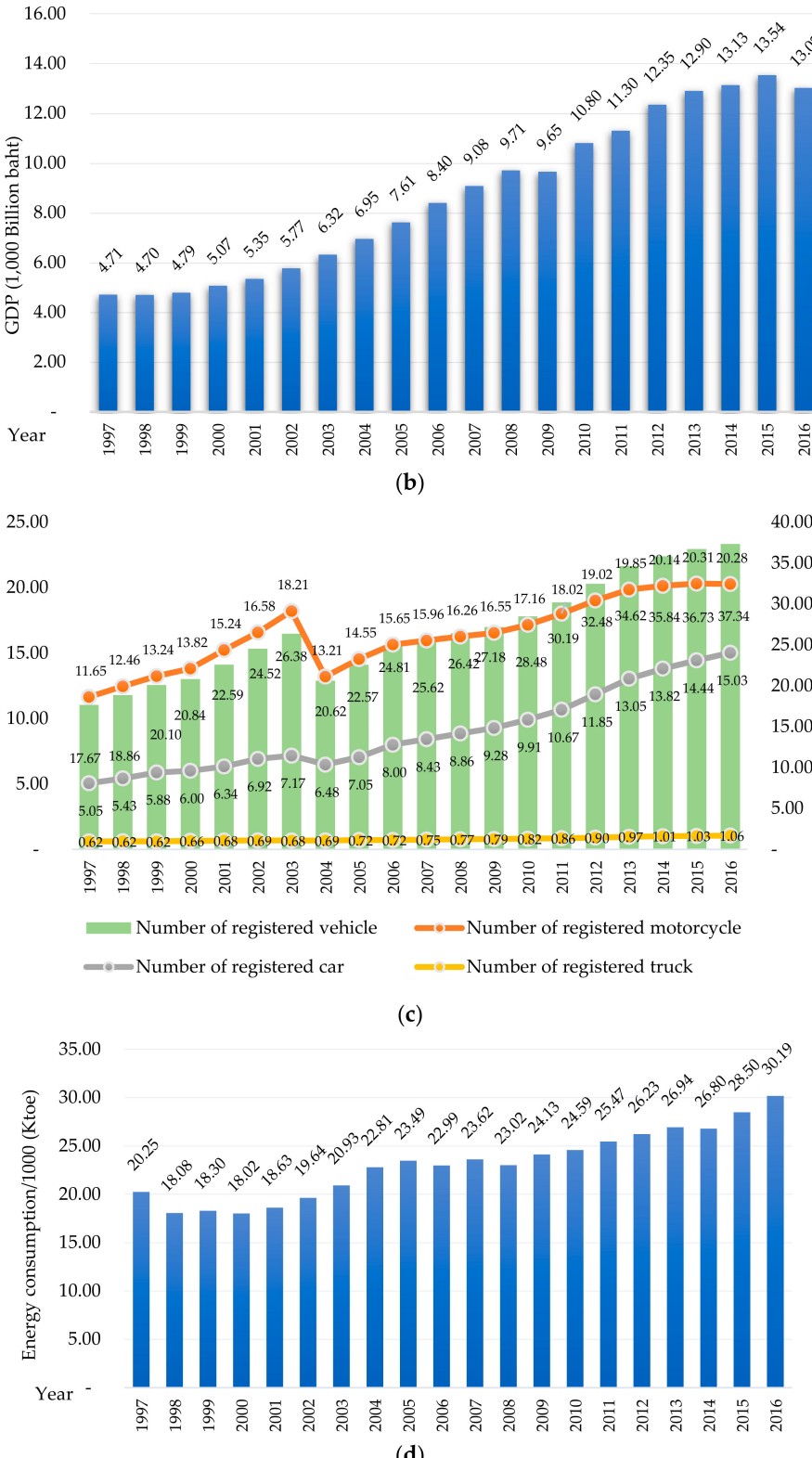

**Figure 2.** The statistical data trends for all components considered in this study: (**a**) Death rate from road accidents per 100,000 population [6], (**b**) Gross domestic product (GDP) statistics from the Bank of Thailand [22], (**c**) Number of registered vehicles statistics (million vehicles) per the Department of Land Transport [23], and (**d**) Energy consumption for transport sector statistics [24].

### 2.3. Data Analysis

The data analysis included four techniques: Time-series analysis, curve estimation, multiple regression analysis, and path analysis. Individual techniques were used differently during each component of our analysis.

#### 2.3.1. Time-Series

Model Specification

Numerous techniques have been proposed for time-series analysis, such as moving average (MA), exponential smoothing (ES), double exponential smoothing model (DES), trend analysis linear regression method (LR), and Winter's method.

Time-series analysis was conducted in our study using the exponential smoothing technique, a technique that values more recent information more highly. Data importance decreases according to the temporal distance of data from the present [25]. The equation is as follows:

$$F_t = aA_{t-1} + (1 - a)F_{t-1}, \tag{1}$$

where $F_{t-1}$ is the value of the prediction during the period prior to forecast phase 1, $A_{t-1}$ is the real value during the period prior to the forecast phase 1, and the smoothing coefficient a takes a value between 0 and 1. The closer the value to 0, the lower its weight [26].

Model Fitting and Validation

We compared the effectiveness of the forecasting model by evaluating the forecast error from a training data set. The lowest mean absolute percent error (MAPE) was used as the criteria for selecting the most effective forecasting model.

#### 2.3.2. Curve Estimation

SPSS 18.0 software (SPSS Inc., Chicago, IL, USA) was used to conduct curve estimation. The independent variable was time (years), with road death rates per 100,000 population calculated using 10 models consisting of linear, logarithmic, inverse, quadratic, cubic, compound, power, S, growth, and exponential models. The equations of the models are as follows, respectively:

$$E(y)_t = b_0 + (b_1 \times T), \tag{2}$$

$$E(y)_t = b_0 + (b_1 \times (\ln(T)), \tag{3}$$

$$E(y)_t = b_0 + (b_1/T), \tag{4}$$

$$E(y)_t = b_0 + (b_1 \times T) + (b_2 \times T^2), \tag{5}$$

$$E(y)_t = b_0 + (b_1 \times T) + (b_2 \times T^2) + (b_3 \times T^3), \tag{6}$$

$$E(y)_t = b_0 \times (b_1)^T, \tag{7}$$

$$E(y)_t = b_0 \times T^b{}_1, \tag{8}$$

$$E(y)_t = \exp[b_0 + b_1/T], \tag{9}$$

$$E(y)_t = \exp(b_0 + (b_1{}^*T)), \tag{10}$$

$$E(y)_t = b_0 \times (\exp^{b_1{}^*T}) \tag{11}$$

where $E(y)_t$ is the predicted value of death rate from road accidents, T is time (T = 1, 2, 3, . . . , 20 years), $b_0$ is a constant (or y-intercept), and $b_1$–$b_3$ are coefficients.

### 2.3.3. Multiple Regression Analysis

Multiple regression analysis was used to determine the relationships between the dependent variables Y to discover which relationships were linear. The following equation shows the relationships between Y and $X_1, X_2, \ldots, X_k$ [27]:

$$\Upsilon = \beta_0 + \beta_1 X_1 + \beta_2 X_2 + \ldots + \beta_k X_k + e, \tag{12}$$

where Y is the dependent variable, $X_1$–$X_k$. are the independent variables, $B_0$ is a constant (or y-intercept), and $B_1$–$B_k$ are the coefficients of the respective variables (loading or partial slopes), which are used to describe the change in the Y value when the X value changes. [27].

Four model specification techniques were used to select the variables in a regression model: (1) All possible regression, where all the independent variables were used in the equation; (2) forward selection, where only one independent variable was used at a time in the model, and the process was repeated until no independent variables were remaining that could explains the variation of the dependent variable; (3) backward elimination, where independent variables are removed to determine which variables only slightly explain the dependent variable, by removing only one variable by the time; and (4) stepwise regression, where only one independent variable is selected by considering the time, similar to forward selection, but in this technique, the independent variable may be removed later, depending on the specified significance level. The equation used was as follows:

$$Y = \beta_0 + \beta_1 \text{VEH\_MOTORCYCLE} + \beta_2 \text{VEH\_CAR} + \beta_3 \text{VEH\_TRUCK} + \beta_4 \text{GDP} + \beta_5 \text{EN\_TRANSPORT}, \tag{13}$$

where Y is the death rate per 100,000 population, VEH_MOTORCYCLE is the number of registered motorcycles, VEH_CAR is the number of registered cars, VEH_TRUCK is the number of registered trucks, GDP is the gross domestic product (billion Baht), and EN_TRANSPORT is the energy consumption (Ktoe).

### 2.3.4. Path Analysis

This technique delineates all the relationships involved in a structural equation model (SEM). Path analysis identifies the details of bivariate relationships between two variables and the weighting of values connected to these points. The researcher determines the number and type of variables and relative paths [28]. The statistics, based on the regression analysis, are used to examine the independent variables that directly and indirectly influence the dependent variable. This describes the relationships between the variables [29]. We used Mplus 7.2 software [30] to conduct the analysis.

Model specification is the procedure used to develop a structural equation model, which is necessary to review concepts, theory, and related studies to form an assumption. A path diagram indicates the relationships between variables, and a structural equation model is developed by following the assumptions in terms of the path diagram and a particular model using a variance–covariance matrix. The appropriated particular model is the model which is supposed to explain the relationships between variables reasonably and consistently with empirical data.

In model estimation, Mplus 7.2 software [30] provided estimated parameters of the model according to the specified values of the model. Several techniques can be used to estimate parameters, for instance, maximum likelihood (ML), generalized least squares (GLS), and generally weighted least squares (WLS). In this study, we considered ML, which has been widely applied and suitable for interval and ordinal scale data. ML is parameter estimation conducted under the assumption that the observed variables are multivariate, normally distributed, and the samples must be independent; that is, they follow a perfect normal distribution (skewness (Sk) ≤ 3; Kurtosis index (Ku) ≤ 10) [31].

Goodness of fit is used to consider the consistency of an SEM developed from empirical data by examining the indices of the model to test the model fit statistical criteria, where the chi-square

value/degrees of freedom (df) < 3 [31], root mean square error (RMSEA) < 0.07 [32], the comparative fit index (CFI) ≥ 0.90 [33], the Tucker–Lewis index (TLI) ≥ 0.80 [34,35], and the SRMR < 0.08 [33].

### 2.3.5. Evaluating Model Efficiency

The MAPE [29] was used to evaluate model efficiency with the following formula:

$$\text{MAPE} = \frac{1}{n}\sum_{i=1}^{n}\left|\frac{F_i - O_i}{O_i}\right| \times 100, \tag{14}$$

where $F_i$ is the predictive value acquired from each year, $O_i$ is the actual value that occurred each year, and $n = 20$ years.

## 3. Results

### *3.1. Statistical Results*

#### 3.1.1. Time-Series Data Analysis

Exponential smoothing was used to analyze the data, and Holt's linear trend technique using two constants was used to establish the priority of data, with data importance gradually decreasing depending on the distance of the data from the present. This technique was used to determine a linear trend data without seasonal influences (Figure 2a). This resulted in a MAPE value of 8.1%.

#### 3.1.2. Curve Estimation Technique

The results of the curve estimation analysis in Table 2 show that the three most accurate models were the cubic model $E(y)_t = 18.262 + 1.487T - 0.189T^2 + 0.005T^3$, the quadratic model $E(y)_t = 20.772+0.207T-0.040T^2$, and the linear model $E(y)_t = 23.879-0.641T$ with adjusted $R^2$ values of 0.813, 0.794, and 0.724, respectively.

**Table 2.** Curve estimate model. R2, determination coefficient; SE, standard error.

| Model | $R^2$ | Adjusted $R^2$ | SE of the Estimate | $F$ | Equation |
|-------|-------|----------------|--------------------|-----|----------|
| Linear | 0.738 | 0.724 | 2.318 | 50.815 | $E(y)_t = 23.879 - 0.641T$ |
| Logarithmic | 0.511 | 0.483 | 3.170 | 18.785 | $E(y)_t = 25.361 - 3.878\ln(T)$ |
| Inverse | 0.249 | 0.208 | 3.926 | 5.981 | $E(y)_t = 15.379 + (9.856/T)$ |
| Quadratic | 0.816 | 0.794 | 2.001 | 37.638 | $E(y)_t = 20.772 + 0.207T - 0.040T^2$ |
| Cubic | 0.842 | 0.813 | 1.908 | 28.511 | $E(y)_t = 18.262 + 1.487T - 0.189T^2 + 0.005T^3$ |
| Compound | 0.729 | 0.714 | 0.152 | 48.518 | $E(y)_t = 25.494(0.960)^T$ |
| Power | 0.489 | 0.460 | 0.210 | 17.210 | $E(y)_t = 27.813T^{-0.245}$ |
| S | 0.223 | 0.180 | 0.258 | 5.179 | $E(y)_t = \exp[2.698 + (0.603/T)]$ |
| Growth | 0.729 | 0.714 | 0.152 | 48.518 | $E(y)_t = \exp[3.238 - (0.041T)]$ |
| Exponential | 0.729 | 0.714 | 0.152 | 48.518 | $E(y)_t = 25.494\exp^{-0.041T}$ |

#### 3.1.3. Multiple Regression Analysis

We multiple regression analysis was used to analyze the results of the models, with the death rate from road accidents as the dependent variable, and independent variables, including the number of registered vehicles, the GDP, and energy consumption. The results, which are shown in Table 3 are as follows:

Model 1: GDP and the energy consumption of the transport sector affected the death rate from road accidents ($p < 0.05$); however, the coefficient values were very low, at $-2.393 \times 10^{-6}$ ($p < 0.001$), and 0.001 ($p < 0.05$), respectively. The model accuracy value was 0.842 (84.2%) and $F$-test value was 51.814.

Model 2: Factors were adjusted for population. We found that the number of registered cars over the population, and the energy consumption of the transport sector over the population, affected the

death rate from road accidents ($p < 0.05$). However, the coefficient values remained very low, at −0.127 ($p < 0.001$) and 0.424 ($p < 0.001$), respectively. The model accuracy value was 0.853 (85.3%), and the *F*-test value was 55.915.

Model 3: Factors were adjusted for the number of registered vehicles. We found that the number of registered cars over the number of registered vehicles, number of registered trucks over the number of registered vehicles, and the energy consumption of the transport sector over the number of registered vehicles affected the death rate from road accidents ($p < 0.05$). However, the coefficient values were still very low, with the coefficient values equal to 0.081 ($p < 0.001$), −0.7329 ($p < 0.05$), and 0.260 ($p < 0.001$), respectively. The model accuracy was = 0.888 (88.8%) and *F*-test value was 51.144.

**Table 3.** Results of multiple regression analysis.

| Variable | Model 1 | | |
| --- | --- | --- | --- |
| | **(Y = Death Rate/100,000 Population)** | | |
| | **B** | ***t*-Statistic** | ***p*-Value** |
| VEH_MOTORCYCLE | - | - | - |
| VEH_CAR | - | - | - |
| VEH_TRUCK | - | - | - |
| GDP | $-2.393 \times 10^{-6}$ | −5.551 | <0.001 ** |
| EN_TRANSPORT | 0.001 | 2.839 | <0.05 * |
| Constant | 12.838 | 2.370 | <0.05 * |
| *F*-test | 51.814 | | |
| Adjusted $R^2$ | 0.842 | | |
| **Variables** | **Model 2** | | |
| | **(Y = Death Rate/100,000 Population)** | | |
| | **B** | ***t*-Statistic** | ***p*-Value** |
| VEH_MOTORCYCLE/1,000 population | - | - | - |
| VEH_CAR/1,000 population | −0.127 | −6.405 | <0.001 ** |
| VEH_TRUCK/1,000 population | - | - | - |
| GDP/1,000 population | - | - | - |
| EN_TRANSPORT/100,000 population | 0.424 | 2.265 | <0.001 ** |
| Constant | 19.632 | 4.399 | <0.001 ** |
| *F*-test | 55.915 | | |
| Adjusted $R^2$ | 0.853 | | |
| **Variables** | **Model 3** | | |
| | **(Death Rate/100,000 Population)** | | |
| | **B** | ***t*-Statistic** | ***p*-Value** |
| VEH_MOTORCYCLE/1,000 VEH_TOTAL | - | - | - |
| VEH_CAR/1,000 VEH_TOTAL | −0.081 | −8.769 | <0.001 ** |
| VEH_TRUCK/1,000 VEH_TOTAL | −0.723 | −2.266 | <0.05 * |
| GDP/1,000 VEH_TOTAL | - | - | - |
| EN_TRANSPORT/1,000 VEH_TOTAL | 0.260 | 4.006 | <0.001 ** |
| Constant | 42.096 | 5.903 | <0.001 ** |
| *F*-test | 51.144 | | |
| Adjusted $R^2$ | 0.888 | | |

Note: VEH_TOTAL, number of vehicles registered; VEH_MOTORCYCLE, number of motorcycles registered; VEH_CAR, number of cars registered; VEH_TRUC, number of trucks registered; GDP, gross domestic product (billion baht); EN_TRANSPORT, energy consumption (Ktoe).

## 3.1.4. Path Analysis

The path analysis results showed that the model results had the following goodness-of-fit values: Chi-square = 17.706, df = 3, $p = 0.0005$, RMSEA = 0.495, CFI = 0.884, TLI = 0.536, and standardized root mean square residual (SRMR) = 0.078. This indicated that the model passed the goodness-of-fit

statistical criteria, where the chi-square/df < 3 [31], RMSEA < 0.07 [32], CFI ≥ 0.90 [33], TLI ≥ 0.80 [34,35], and SRMR < 0.08 [33]. We found that only the SRMR was according to the condition.

According to Table 4, the factors that directly affected death rate from road accidents were GDP, the energy consumption of the transport sector, and number of registered cars, with unstandardized factor loadings and standardized factor loadings of −0.106 (−0.193), 0.834 (0.992), and −0.120 (−1.485) at *p*-values of <0.05 (<0.05), <0.001 (<0.001), and <0.05 (<0.05), respectively. The confidence intervals between the independent variables and dependent variables were 86.6% ($R^2 = 0.866$); however, the transport sector energy consumption factor had an influence on the GDP, thus, indirectly affecting the number of road deaths, as seen in the path analysis model results in Figure 3.

**Table 4.** Path analysis model results.

| Relationship | Model Stat | | | |
|---|---|---|---|---|
| | **B (β)** | **SE** | ***t*-Statistic** | ***p*-Value** |
| GDP → FA_RATE | −0.106 (−0.193) | 0.033 (0.352) | −3.245 (−3.390) | <0.05 *(<0.05 *) |
| EN_TRANSPORT → FA_RATE | 0.834 (0.992) | 0.214 (0.274) | 3.896 (3.615) | <0.001 **(<0.001 **) |
| VEH_MOTORCYCLE → FA_RATE | 0.041 (0.432) | 0.022 (0.233) | 1.881 (1.857) | 0.060 (0.063) |
| VEH_CAR → FA_RATE | −0.120 (−1.485) | 0.060 (0.747) | −2.021 (−1.989) | <0.05 *(<0.05 *) |
| VEH_TRUCK → FA_RATE | 0.690 (0.396) | 1.073 (0.614) | 0.643 (0.645) | 0.520 (0.519) |
| EN_TRANSPORT → GDP | 8.898 (0.938) | 0.734 (0.027) | 12.120 (35.012) | <0.001 **(<0.001 **) |
| VEH_MOTORCYCLE→ EN_TRANSPORT | −0.044 (−0.388) | 0.027 (0.240) | −1.623 (−1.615) | 0.105 (0.106) |
| VEH_CAR → EN_TRANSPORT | 0.137 (1.420) | 0.074 (0.752) | 1.863 (1.888) | 0.062 (0.059) |
| VEH_TRUCK → EN_TRANSPORT | −0.345 (−0.166) | 1.411 (0.680) | −0.0244 (−0.244) | 0.807 (0.807) |
| **Intercept** | | | | |
| FA_RATE | −0.799 (−0.194) | 10.513 (2.554) | −0.076 (−0.076) | 0.939 (0.939) |
| GDP | −187.266 (−4.035) | 27.035 (0.639) | −6.927 (−6.317) | <0.001 **(0.001 **) |
| EN_TRANSPORT | −0.345 (6.629) | 10.361 (0.680) | 3.131 (2.783) | <0.05 *(<0.05 *) |
| **Residual variances** | | | | |
| FA_RATE | −2.262 (0.134) | 0.715 (0.055) | 3.162 (2.430) | <0.05 *(<0.05 *) |
| GDP | 258.150 (0.120) | 81.634 (0.050) | 3.162 (2.383) | <0.05 *(<0.05 *) |
| EN_TRANSPORT | 4.040 (0.169) | 10.361 (0.069) | 3.162 (2.452) | <0.05 *(<0.05 *) |
| $R^2$ | | | | |
| FA_RATE | 0.866 | 0.055 | 15.739 | <0.001 ** |
| GDP | 0.880 | 0.050 | 17.506 | <0.001 ** |
| EN_TRANSPORT | 0.831 | 0.069 | 12.087 | <0.001 ** |

*, ** Denotes significance at 0.05, 0.001 level; FA_RATE, death rate/100,000 population.

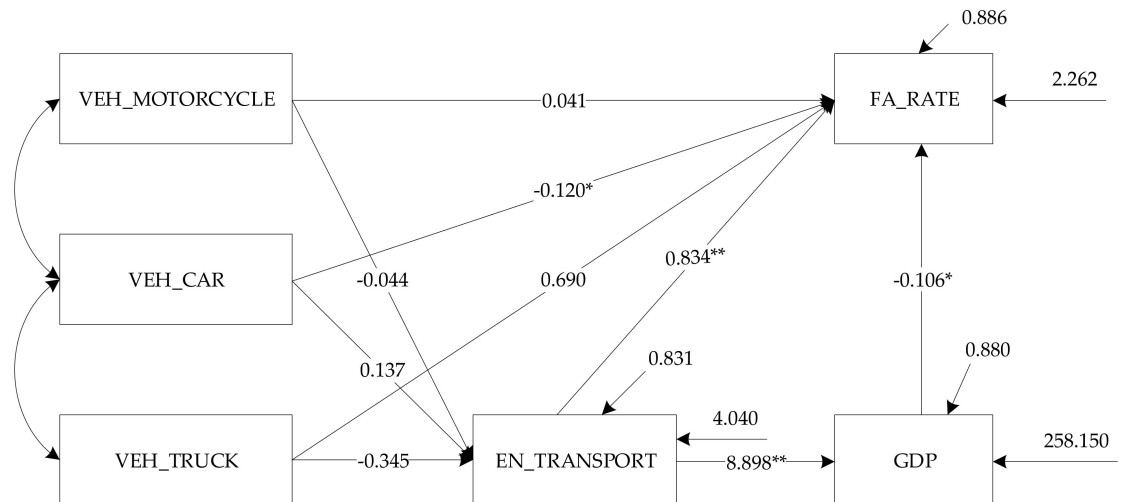

Goodness-of-fit statistics : χ²=17.706, df = 3, P=0.0005, RMSEA =0.495, CFI = 0.884, TLI = 0.536, SRMR=0.078.

*, ** Denotes significance at 0.05, 0.001 level

**Figure 3.** Results of path analysis.

### 3.2. Comparison of Model Performance

We constructed eight models to predict the number of fatalities from road accidents using different predictive techniques to identify the most effective model using MAPE (Equation (15)). Table 5 lists the results of our comparison; we found that multiple regression linear model 3 had the lowest MAPE value (6.4%), followed by the time-series model analysis with exponential smoothing, path analysis, multiple regression linear model 2, curve estimation (quadratic), curve estimation (cubic), curve estimation (linear), and multiple regression linear model 1, with MAPE values of 8.1%, 8.4%, 9.5%, 10.2% 11.2%, 12.6%, and 12.8%, respectively.

**Table 5.** Comparison of the mean absolute percentage error (MAPE) of different methods.

| | Model | Year | | | | | Value | |
|---|---|---|---|---|---|---|---|---|
| | | 1997 | 2001 | 2006 | 2011 | 2016 | MAE * | MAPE * |
| 1 | Time-series (exponential smoothing) | 22.53 | 18.68 | 20.05 | 11.78 | 9.09 | 1.62 | 8.1 |
| 2 | Curve estimate (cubic) | 19.57 | 21.60 | 19.23 | 14.92 | 12.40 | 2.23 | 11.2 |
| 3 | Curve estimate (quadratic) | 20.94 | 20.81 | 18.84 | 14.88 | 8.91 | 2.04 | 10.2 |
| 4 | Curve estimate (linear) | 23.24 | 20.67 | 17.47 | 14.26 | 11.06 | 2.52 | 12.6 |
| 5 | Multiple regression linear model 1 | 21.82 | 18.68 | 15.72 | 11.26 | 11.87 | 2.56 | 12.8 |
| 6 | Multiple regression linear model 2 | 23.20 | 19.08 | 18.44 | 14.19 | 7.67 | 1.91 | 9.5 |
| 7 | Multiple regression linear model 3 | 23.65 | 19.26 | 19.13 | 14.97 | 10.08 | 1.28 | 6.4 |
| 8 | Path analysis | 24.40 | 21.13 | 19.17 | 15.16 | 11.13 | 1.68 | 8.4 |

MAE, mean absolute error; MAPE, mean absolute percentage errors; *, average MAPE 1997–2016.

### 3.3. Forecasting the Death Rate from Road Accidents per 100,000 Population

As displayed in Table 6, the predictions based upon the compared models used additional data (i.e., GDP, the number of registered vehicles (motorcycles, cars, and trucks), and the transport sector energy consumption). The second-best forecasting technique included these related factors when conducting time-series analysis with the exponential smoothing technique. As shown in Table 7, a 10-year forecast was generated. The results of our prediction showed that the time-series model (exponential smoothing), curve estimation (quadratic), curve estimation (linear), multiple regression 2–3, and path analysis indicated a declining direction, reflecting the fatality statistics from the National Police Bureau (Figure 2). However, some models predicted that the reduction was close to zero (no decline in the death rate from road accidents) and the multiple regression linear 1 and curve estimation (cubic) models predicted increasing tendencies. Figure 4 presents a comparison of the predictions using the various techniques.

**Table 6.** Predictive results, including the variables and prediction to be used in the models.

| Parameter | Year | | | | | |
|---|---|---|---|---|---|---|
| | 2020 | 2022 | 2024 | 2026 | 2028 | 2030 |
| Population ($10^6$) | 66.84 | 67.30 | 67.77 | 68.24 | 68.71 | 69.17 |
| GDP (1012 baht/1000 population); 1997 constant price | 253.17 | 269.79 | 286.41 | 303.04 | 319.66 | 336.28 |
| Motorcycle registration /1000 population | 333.62 | 344.74 | 355.85 | 366.97 | 378.08 | 389.20 |
| Car registration /1000 population | 267.55 | 287.37 | 307.19 | 327.01 | 346.84 | 366.66 |
| Truck registration /1000 population | 17.72 | 18.57 | 19.42 | 20.27 | 21.12 | 21.97 |
| Energy consumption of the transportation sector (Ktoe/100,000 population) | 48.91 | 50.47 | 52.03 | 53.59 | 55.15 | 56.72 |
| Motorcycle/1000 Total Vehicles registered | 516.00 | 502.12 | 488.25 | 474.37 | 460.49 | 446.61 |
| Car/1000 Total Vehicles registered | 428.52 | 441.85 | 455.19 | 468.52 | 481.85 | 495.19 |
| Truck/ 1000 Total Vehicles registered | 26.45 | 25.97 | 25.50 | 25.02 | 24.54 | 24.07 |
| GDP /1000 Total Vehicles registered; 1997 constant price | 425.16 | 443.38 | 461.59 | 479.81 | 498.03 | 516.24 |
| Energy consumption of the transportation sector (Ktoe)/1000 Total Vehicles registered | 76.64 | 74.53 | 72.42 | 70.32 | 68.21 | 66.10 |

**Table 7.** Death rate from road accidents per 100,000 population as predicted by the models used.

| Model | | Year | | | | | |
|---|---|---|---|---|---|---|---|
| | | **2020** | **2022** | **2024** | **2026** | **2028** | **2030** |
| 1 | Time-series (exponential smoothing) | 9.82 | 8.54 | 7.26 | 5.98 | 4.70 | 3.42 |
| 2 | Curve estimate (cubic) | 14.20 | 17.00 | 21.50 | 27.80 | 36.20 | 46.90 |
| 3 | Curve estimate (quadratic) | 2.70 | - | - | - | - | - |
| 4 | Curve estimate (linear) | 8.50 | 7.21 | 5.93 | 4.65 | 3.37 | 2.09 |
| 5 | Multiple regression linear model 1 | 12.20 | 13.30 | 14.40 | 15.50 | 16.60 | 17.7 |
| 6 | Multiple regression linear model 2 | 6.39 | 4.54 | 2.68 | 0.82 | - | - |
| 7 | Multiple regression linear model 3 | 8.19 | 6.91 | 5.62 | 4.34 | 3.06 | 1.77 |
| 8 | Path analysis | 7.75 | 5.96 | 4.16 | 2.36 | 0.56 | - |

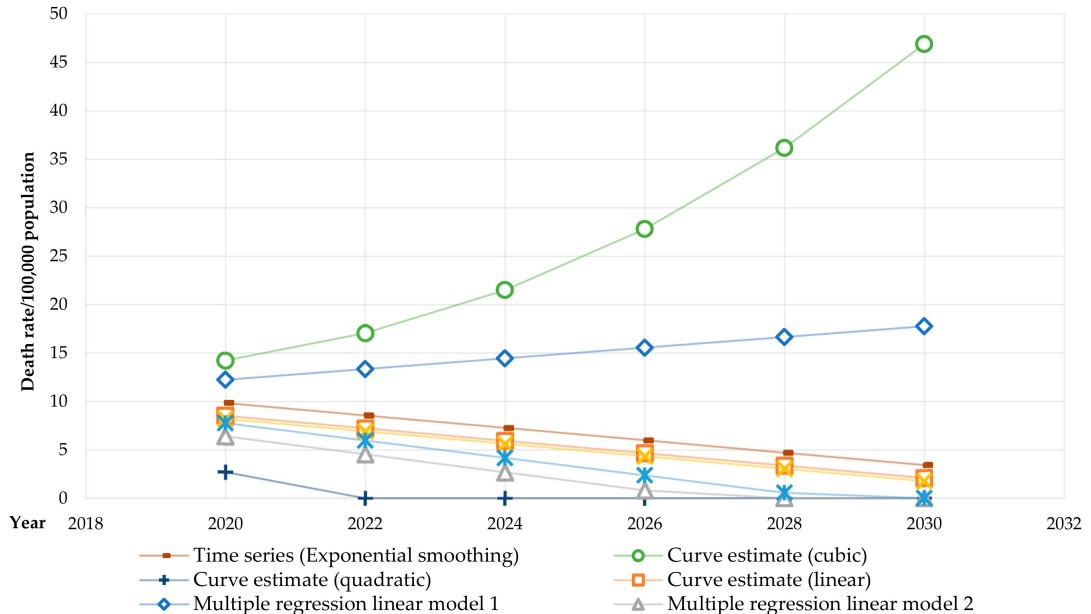

**Figure 4.** Comparison of prediction results using different techniques.

## 4. Discussion

Before Thailand was ranked first in road accident deaths per 100,000 population in 2017, the WHO had already ranked Thailand's road accident situation as the world's second-worst in 2015. This ranking created the unpleasant image that travel in Thailand is unsafe. The organizations and departments involved have recognized the need to prioritize measures to decrease the worsening road safety situation, including law enforcement, education campaigns for schools, advertising media, and increasing the training hours required to obtain a new driver's license or renew a license, as well as using technical engineering solutions. Funding for research has been provided to investigate and identify solutions to this problem in an atmosphere of economic growth focusing on travel, accident risks, and into solutions for reducing the number of accidents and deaths. Among the measures mentioned above, the National Police Bureau statistics indicated that the death rate has been decreasing; however, in 2016, the rate actually increased (Figure 2a).

The GDP, the number of registered vehicles, and transport sector energy consumption are likely to increase in the future (Figure 2b–d). Therefore, in this study, we analyzed statistical data to forecast the death rate from road accidents by a time-series model using exponential smoothing, curve estimation, multiple linear regression, and path analysis, using official Thai statistical data collected over the past 20 years. The research results are as follows:

(1) The time-series model using exponential smoothing is suitable for predicting the death rate from road accidents. Time-series techniques have also been used to forecast accidents by

ARIMA [9–12,16]. However, the Thai data set was not suitable for the ARIMA technique. The exponential smoothing technique yielded MAE and MAPE values of 1.627 and 8.1%, respectively.

(2)    Curve estimation with cubic, quadratic, and linear patterns were the three models with the highest $R^2$ values.

(3)    Multiple regression linear model 1 found that GDP was a good economic indicator, in agreement with a previous report by Dadashova et al. [16], and that the transport sector energy consumption level affected the death rate from road accidents, in agreement with reported results García-Ferrer et al. [12]; the number of registered vehicles (motorcycle, cars, and trucks) had no effect.

(4)    Using multiple regression linear model 2 (where the proportion of various factors was adjusted by population), we found that the number of registered vehicles and transport sector energy consumption affected the death rate from road accidents, whereas, the other factors had no effect.

(5)    Using multiple regression linear model 3 (where the proportion of factors was adjusted for the number of registered vehicles), we found that the number of registered vehicles, number of registered trucks, and the amount of energy consumed by the transport sector affected the death rate from road accidents, whereas, the other factors had no effect.

(6)    The path analysis model showed that GDP, energy consumption, and the number of registered vehicles were factors that directly influenced the road death rate. The amount of energy consumed by the transport sector was a factor influenced by the GDP, which indirectly affected the number of road deaths.

The effectiveness of the first three models with the lowest MAPE were multiple regression linear model 3, the time-series (exponential smoothing) model, and the path analysis model, with MAPE values of 6.4%, 8.1%, and 8.4%, respectively.

When the models were used to predict the death rate from road accidents, we found that the time-series (exponential smoothing), curve estimation (quadratic), curve estimation (linear), multiple regression 2 and 3, and path analysis models forecasted decreasing fatal accident trends, which supports the data on the direction of the death rates provided by the Royal Thai Police [6]. We found that the multiple regression linear 1 and curve estimation (cubic) models generated forecasts that contrasted the trends observed in the data provided by the National Police Bureau, whereas, the curve estimation (quadratic), multiple regression linear 1, and path analysis models predicted a zero value, and thus, are not suitable for long-term forecasting. Only the time-series (exponential smoothing), curve estimation (linear), and multiple regression linear 3 models generated predictions that were consistent with the trends present in the original statistical data.

ARIMA models have been applied to accident forecasting [9]. However, Thailand's data were not appropriate for applying an ARIMA model, but were suitable for exponential smoothing, and hence, could be used in forecasting. The economic growth data that were used in forecasting by applying multiple regression linear and path analysis were GDP, the energy consumption of the transport sector, and the number of registered vehicles (motorcycles, cars, and trucks).

According to our data, none of the models were found to be suitable for predicting the death rate from road accidents 10 years in advance. The road death rates will feasibly decline over the long time period of over 10 years and the various measures implemented. The economic and transportation factors considered, which reflect the economic growth of the country, had both direct and indirect effects on the road death rates. If considered in -depth, some data may be useful for informing government policy-making and for designing preventive measures to reduce the causes of accidents, especially the number of registered cars on the roads, which is directly related to the number of accidents. In addition to personal and environmental factors, the appropriate control of the rate of vehicle occupancy should be considered. The legal driving age should be reviewed, along with knowledge of traffic regulations and proven experience in safe driving, when applying for a driver's license.

Appropriate policies are required to reduce fatal accidents in the public sector. Due to the mixed traffic road conditions in Thailand, trucks and other large vehicles share roadways with other small- or medium-sized vehicles, which may cause dangerous situations and accidents. The public sector

should implement policies to rigorously control the driving speed, covering all types of vehicles and providing exclusive lanes for freight vehicles. Principally, these policies may help drive Thailand's economic growth, consistent with the results of multiple linear regression model 1, in which positive growth of the economic factor GDP, as an overall gross product of the nation, indicated increased economic activity (e.g., import and export, and generation of jobs and income).

The public sector must create policies related to the control of the possession of vehicles, including stricter measures, such as requiring declaring a driver's license, and consideration of traffic violation history and accident history, to possess a vehicle. This policy would be consistent with the results of multiple linear regression models 2 and 3 and the path analysis model. Energy consumption in the transportation sector was found to be connected to the recent number of registered vehicles in Thailand, which has been increasing.

Other factors affecting these policies could be investigated in terms of budgets for solving the accident problems considered in this study for mitigating and preventing accidents or deaths; for example, by providing knowledge and understanding of accidents through public relations by community leaders or organizations, or providing a driver's license.

## 5. Limitations and Future Work

Model analysis involves forecasting limitations, which potentially result in the misleading prediction of trends. When attempting to forecast road accident death rates, other factors must be considered, including law enforcement measures, such as those on speed limits, drunk driving, helmet wearing, seat belt use, and phone use while driving and other distracted driving behaviors, in addition to public transport use, transport infrastructure developments, and other economic and social issues, which have yet to be analyzed systematically (lane markings, lighting, road markers, signage, intersections, warnings).

The data that were used in our analysis, due to Thailand incompletely collecting data, according to the plan that was set 20 years ago, led to the lack of many types of data in the analysis. Thailand's accident data has several databases, which affected the consistency of the data.

**Author Contributions:** Conceptualization, S.J.; Data curation, S.U.; Formal analysis, S.J. and S.U.; Funding acquisition, S.J.; Methodology, S.J.; Supervision, V.R. All authors have read and agreed to the published version of the manuscript.

**Funding:** This research was funded by the Suranaree University of Technology Research and Development Fund grant number IRD7-704-61-12-11 and the APC was funded by Suranaree University of Technology.

**Acknowledgments:** The authors would like to thanks the Suranaree University of Technology Research and Development Fund.

**Conflicts of Interest:** The authors declare no conflict of interest.

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
