# Peer review of "Forecasting Road Traffic Deaths in Thailand: Applications of Time-Series, Curve Estimation, Multiple Linear Regression, and Path Analysis Models"

_sustainability, doi:10.3390/su12010395_

Round 1
Reviewer 1 Report
Comments and Suggestions for Authors
This paper deals with forecasting road fatalities in Thailand using four approaches namely time-series, curve estimation, multiple linear regression, and path analysis. There are major and fundamental issues with this paper.
1- First, the language of the paper needs major edition. The write-up is full of grammatical mistakes, jargon language and inconsistent arguments which have largely prevented the understandability of the paper. In addition, the employed terminology is not consistent with the road safety profession. I am presenting only a few examples of this comment:
· Inconsistent terminology: use of terms “death” or “mortality” instead of “fatality”
· Jargon language: “worse road safety situation”
· Vague phrases: “the trend has been steady” (the trend in what? and where?)
· Incorrect statements: “road death rate was 22.75 in 1997 and increased to 20.89 in 2002.”; increased or decreased? or “we discovered that the number of road deaths increased”; number or rate?
· The abstract is disorganized, confusing and vague. It is not self-explanatory and it does not present any informative statement about the study.
The manuscript has clearly been written without sufficient care. The authors are advised to undertake major edition with respect to the language of the paper. After all, a major part of any scientific research is its language.
2- The manuscript has no story telling. In particular, the paper lacks the three fundamental elements for any scientific research: 1) research gaps, 2) research questions, and 3) research objectives. What is the research gap that this study aims to cover? What is the research question with respect to that research gap? What is the research objective? (The objective cannot be “to use time-series, curve estimation, multiple linear regression and path analysis to forecast the road accident death rate”. This statement rather presents some tasks to achieve the objective(s).
3- The paper lacks a proper literature review. Although there is too much information regarding the context of the study, this information has been presented without pursuing a goal. The authors are reminded that the main purpose of literature review in a paper is to highlight the research gaps in the literature and thus the motive to undertake the study.
4- The study has analysed 20 years of data. While no information and descriptive statistics have been presented about the data (which is highly needed and must be there in the paper), the use of such long-period observations has fundamental pitfalls, particularly for regression analysis and path analysis. The explanatory variables in these models, obtained from such data, are highly prone to changes which can negatively affect the accuracy of these models.
5- The details and the structure of the employed methods have not been thoroughly presented. I can only see the generic model specification for the time series, regression and path analysis (and nothing for the curve estimation). Model specifications, estimation methods, distributional assumptions, etc. must be presented in detail, for all of the methods.
Reviewer 2 Report
Comments and Suggestions for Authors
The authors aim study is to forecast death rate on road accidents in Thailand by using several approaches: time-series, curve estimation, multiple linear regression and path analysis. For that purpose, they have used existing official data from the last 20 years.
This reviewer has thoroughly read the manuscript. It is well written in general terms, albeit some English proofread is mandatory to polish grammar and some strange expressions. The methods and results are sufficiently described, but it is also recommended to revise the structure of the manuscript to make it more readable and easy to follow.
There is some confusion between curve fit models mentioned in Section 2 and multiple regression models on Section 3. Use two different words or acronyms to clearly distinguish these two concepts. Define the models 1 to 3 properly in the methodology section.
Please, check carefully the contents of tables and figures, and correct the multiple grammar mistakes existing on them.
Specific comments (non-exhaustive list):
L19: Define MAPE acronym.
L20: Is really reliable this degree of precision in percentages (0.001% precision)? I recommend to use 1 or 2 decimal places.
L48: Reference [2] would be better placed on L47.
L49 "death" before rate is missing.
Figure 1 and wherever necessary: Include source in the caption, not before it.
L72: Please, rewrite the sentence to clarify it more.
L93: Correct capital A in "Average". Should it be AADT (Annual Average Daily Traffic)?
L101: Move the year period to a previous place in the sentence.
Figures 2-5 should be merged into a single Figure 2 with four letters (a to d) and one single caption. Correct the existing grammar mistakes on them.
L138: Use colon instead of semicolon.
L143, 171: Include reference and version of these software packages.
L175, Eq. 3: Correct typos "? 00"
L195: Sort the model numbers from 1 to 3. Rewrite to avoid redundances (e.g.:Model 2: Model 2 factors...)
Round 2
Reviewer 1 Report
Comments and Suggestions for Authors
Thanks for addressing the comments. There are still a few grammatical mistakes in the language of the paper. It is worth it to do another round of proof reading.
Author Response
Thank you for your review and feedback. The minor grammatical mistakes were revised by MDPI English Editing Team.